# TEXT-DRIVEN ZERO-SHOT DOMAIN ADAPTATION WITH CROSS-MODALITY GRAPH MOTIF MATCHING

## ABSTRACT

Zero-shot domain adaptive semantic adaptation aims to transfer knowledge from a source domain and learn a target segmenter without access to any target domain data. Some existing methods have achieved notable performances by transforming source features to the target domain through language-driven methods. However, these methods often align language features to global image features coarsely resulting in sub-optimal performance. To address the challenges, we propose a graph motif-based adaptation method designed to balance the efficiency and effectiveness of feature alignment. Our approach involves constructing motif structures based on domain-wise image feature distributions. By increasing the angle between language-vision directed edges, we effectively pull visual features toward the language feature center, thereby achieving cross-modality feature alignment. Additionally, we employ relationship-constraint losses, *i.e.,* directional and contrastive losses, to mitigate the mode-collapse during target feature stylization. These relationship-constraint losses help stabilize the learning process and improve the robustness of the adaptation. Extensive experimental results validate the efficacy of our proposed method. The code for this method will be made available.

## 1 INTRODUCTION

In the field of computer vision, semantic segmentation Zhao & Tao (2023); Kang et al. (2020); Chen et al. (2017) has attracted much attention from researchers due to its pivotal role in scene understanding, *e.g.,* autonomous driving. However, in practical scenarios, there often exists a significant gap in data distribution between the training data, which is used to learn the model, and the test data, upon which the model is deployed. Domain gaps may exist in different weather conditions, such as rain, snow, etc., or different light conditions, such as day and night. These domain gaps Deng et al. (2009) can lead to performance degradation when the source-trained model is deployed in the target domain.

Zero-shot domain adaptive semantic segmentation (ZSDA-SS) has been introduced to address the challenges posed by domain gaps. Unlike unsupervised domain adaptation (UDA), ZSDA-SS does not require any target domain data to be involved in the training stage, making it more practical for real-world applications. Some CLIP-based methods Yang et al. (2024) use the language description of the target domain as the guide to transform the source image features to the target domain stylized features, and they allow us to learn the target domain segmenters in a zero-shot paradigm. Though these methods have achieved significant performance enhancements, they still exhibit two primary limitations. Firstly, the existing methods cannot accomplish a proper balance between the computation cost and the graininess of alignment across domains. They represent the image feature via a global feature vector, which is then aligned with the text embedding Fahes et al. (2023). This coarse alignment of compressed image features inevitably leads to the loss of domain-specific information, adversely affecting the effectiveness of feature alignment. Secondly, most existing methods typically achieve cross-modality feature alignment by directly increasing the cosine similarities of each paired image and text sample Kerr et al.; Wang et al. (2022); Xiao et al. (2023). This approach, however, focuses solely on the optimization of a single sample pair, which can cause the shared feature space to lack diversity. This may lead to mode-collapse during feature stylization, as the model becomes overly specialized to specific pairs and loses the ability to handle a broader range of styles and variations effectively Gal et al. (2022). Therefore, it is essential to develop a feature align-

Figure 1: Illustration of our proposed zero-shot domain adaptation method. It transforms the source image features into stylized features under the driven of target text descriptions. It constructs a hybrid cross-modality graph and utilizes the motif matching strategy to achieve cross-domain alignment.

ment method that incorporates multiple sample constraints to improve the diversity and robustness of feature stylization.

To address these challenges, we propose a graph motif-based zero-shot test time adaptation method, as illustrated in Fig. 1. Our method provides an efficient framework that transfers models from the source domain to multiple target domains simultaneously. We adopt the Prompt-driven Instance Normalization (PIN) module from PØDA to transform the source features into target stylized features. Unlike existing methods, we use the mean and variance parameters to estimate the distribution range of the transformed features for each target domain. We leverage the domain-wise target feature distributions along with the text embedding of the domain descriptions to construct a hybrid graph across modalities. Specifically, we define a motif structure that describes the relationships among the extreme features within the distribution range of visual features and the text embedding themselves. By maximizing the angle between the language-vision directed edges within the matched motifs, we guide the visual features to converge around the linguistic feature centers, thus achieving cross-modality feature alignment. Moreover, to prevent the style transformation process from succumbing to mode-collapse, we introduce directional and contrastive losses. These losses act both as constraints and as guidance within the feature space, thereby enhancing the diversity of the transformed target features and improving the overall effectiveness of domain adaptation.

We summarize our contributions as follows:

- We propose a graph motif-based method for the zero-shot domain adaptive semantic segmentation (ZSDA-SS) problem. To the best of our knowledge, this is the first work to address the ZSDA-SS task by matching graph motifs across modalities and domains.

- We introduce directional and contrastive losses to constrain the stylization process and prevent mode-collapse.

- Extensive experiments conducted on benchmark tasks demonstrate that our method achieves state-of-the-art adaptation performance.

## 2 BACKGROUND

### 2.1 UNSUPERVISED DOMAIN ADAPTATION

Unsupervised Domain adaptation (UDA) has been extensively studied for its potential in model generalization and deployment. It has been widely utilized in many files of computer vision, such as image classification He et al. (2016); Sener et al. (2016); Wang & Jiang (2022); Wang et al. (2021), image segmentation Cao et al. (2023); Zhou et al. (2023); Jin et al. (2023); He et al. (2020); Wu et al. (2024), object detection Redmon et al. (2016); Wu et al. (2022c;b); Liu et al. (2023b;c), and image clustering Liu et al. (2022; 2023a). DANN Ganin & Lempitsky (2015) utilizes Generative Adversarial Networks to introduce a Gradient Reversal Layer that addresses domain adaptation challenges effectively. CIGAR Liu et al. (2023b) proposes to transform the image features into graphs

and employ a graph-matching method to realize feature alignment. The concept of One-Shot Unsupervised Domain Adaptation (OSUDA) is further explored by researchers who aim to transfer knowledge using only one example from the target domain. Luo et al. Luo et al. (2020) propose to use a generator to extract style information from images, which helps in reducing overfitting issues in domain adaptation. Wu et al. Wu et al. (2022a) employ a patch-matching method to blend information from target styles to enhance adaptation accuracy. Lengyel et al. Lengyel et al. (2021) propose Zero-Shot Domain Adaptation (ZSDA), a setting in which no target domain data is available throughout the adaptation process PØDA Fahes et al. (2023) proposes to utilize the image encoder of the language-vision model as the backbone to extract image features. They propose a Prompt-driven Instance Normalization to transfer the source features into target domains and fine-tune the target model. ULDA Yang et al. (2024) extends PØDA in a hierarchical manner. They propose to align the transformed target domain features at global, category, and pixel levels. However, it significantly increases the training computational cost and thus limits their practical applications.

## 2.2 VISION-LANGUAGE MODELS

Extracting the relationship between vision and language modalities has been an important research area in recent years Lu et al. (2019); Lee et al. (2009); Tan & Bansal (2019); Gao et al. (2019); Devlin et al. (2018); Dosovitskiy et al. (2020); Liu et al. (2021). CLIP Radford et al. (2021) proposes to employ a contrastive learning method to learn the feature similarities between language-vision sample pairs, setting a foundational precedent for subsequent research in this area. CoOp Zhou et al. (2022b) proposes to use learnable variables to represent the text prompt, instead of using fixed hand-craft prompts. By the learnable prompts, CoOp gains a stronger ability to extract text features. CoCoOp Zhou et al. (2022a) proposes to extract the image features as the condition of text prompts to further enhance the relationship between visual and linguistic features. The CLIP-based model uses a lot of training data so that it contains a wealth of knowledge, which makes it obtain excellent zero-shot image classification capability. This robust knowledge base allows these models to bridge the gap between textual and visual data effectively, catalyzing advancements in text-driven image editing applications such as image stylization. StyleCLIP Patashnik et al. (2021) introduces a latent mapper that aligns the features of an input image with text guidance descriptions. CLIPStyler Kwon & Ye (2022) utilizes text descriptions to define the desired style and employs CLIP to transform the image into the specified style by minimizing the distance between the transformed image and the description text in the shared feature space. It is realized by pulling the distance between the converted image and the description text in the shared feature space. The vision-language pretrained models have been widely employed in many other computer vision fields Kerr et al.; Wang et al. (2022); Xiao et al. (2023).

## 3 PRELIMINARY

### 3.1 PROMPT-DRIVEN ZERO-SHOT DOMAIN ADAPTATION (PØDA)

PØDA Fahes et al. (2023) is a recent method designed to deal with the ZSDA-SS problem using the pretrained vision-language model CLIP. It consists of two steps, *i.e.,* i) the stylization of target domain features, and ii) the fine-tuning of the target segmenter.

The first step of PØDA leverages the text description of the target domain as a prompt to extract language knowledge, which guides the alignment of visual features across domains. Specifically, it introduces a Prompt-driven Instance Normalization (PIN) module to transform the source image features $f_s$ extracted from the CLIP image encoder to the target stylized features $f_{s \to t}$. The stylization process is mathematically formulated as follows:

$$
\begin{aligned}
f_{s \to t} &= \text{PIN}(f_s, \mu, \sigma) \\
&= \sigma\left(\frac{f_s - \mu(f_s)}{\sigma(f_s)}\right) + \mu,
\end{aligned}
\tag{1}
$$

where $\mu$ and $\sigma$ are trainable parameters that represent the style information of target domain features. $\mu(f_s)$ and $\sigma(f_s)$ are channel-wise mean and standard deviation of input source features. By minimizing the following loss function, the PIN module enhances the similarity between $f_{s \to t}$ and

CLIP text embedding TrgEmb, which characterizes the style of a target domain.

$$L_{PIN}(\mathrm{f}_{s \to t}, \mathrm{TrgEmb}) = 1 - \frac{avg(\mathrm{f}_{s \to t}) \cdot \mathrm{TrgEmb}}{\|avg(\mathrm{f}_{s \to t})\| \cdot \|\mathrm{TrgEmb}\|}, \tag{2}$$

where $avg()$ is the average pooling operation to pool the feature map into a vector. After training the PIN module using source images and corresponding descriptions of the target domain, the target style information is encoded into the PIN parameters $\mu$ and $\sigma$. With Eq. 1, we can transform each source image feature to its counterpart in the target domain. In the second step, we obtain the target segmenter by fine-tuning the classification head with these features $\mathrm{f}_{s \to t}$ in the target style and their labels. For further details, please refer to Fahes et al. (2023).

## 3.2 GRAPH MOTIF

Graph theory West et al. (2001) has proven its effectiveness in the field of computer vision, where the concept of graph motifs has been increasingly utilized to enhance structural and relational understanding in image and video analysis. A graph motif, defined as a recurring and significant subgraph pattern within a large graph, represents a form of a higher-order graph that exists between second-order relational distances and complex graph structures. These motifs offer a powerful means to capture intricate relationships and local features prevalent in visual data. For instance, MotifNet Zellers et al. (2018) employs motifs to represent the relationships between semantic nodes and generate scene graphs, while SOMA Li et al. (2023) constructs motifs with category-wise prototypes from different domains and implements cross-domain alignment to address the adaptive open-set object detection problem. A more detailed introduction to the use of graph motifs is presented in Chen et al. (2022); West et al. (2001).

## 4 METHOD

**Problem formulation.** Let $\mathcal{D}_s = \{(x_s, y_s)\}$ represent the source domain, where $x_s \in \mathbb{R}^{h \times w \times 3}$ denotes a source image and $y_s \in \mathbb{R}^{h \times w \times C}$ is the corresponding pixel-wise label map. Here, $h$ and $w$ are the height and width of an image, respectively, and $C$ represents the number of semantic categories. Similarly, $N$ target domains are denoted by $\{\mathcal{D}_t^i\}_{i=1}^N = \{\{(x_t^1, y_t^1)\}, \ldots, \{(x_t^N, y_t^N)\}\}$. The segmenter is defined as $G_{s/t} = E_{img} \odot H_{s/t}$, where $E_{img}$ is the backbone utilizing the frozen pretrained image encoder from CLIP, and $H_{s/t}$ is the segmentation head. Given the pretrained source segmenter $G_s$ and the text description $\{\mathrm{TrgDesc}^i\}_1^N$ of $N$ target domains, our method aims to learn a target semantic segmenter $G_t$ for each target domain driven by its text description.

## 4.1 OVERVIEW

Fig. 2 illustrates the structure of our proposed method, which consists of two processes: stylization of target domains and target segmenter fine-tuning. During the stylization process, we extract the features of source images $f_s = E_{img}(x_s)$ with CLIP image encoder $E_{img}$ and input the descriptions $\{\mathrm{TrgDesc}^i\}_{i=1}^N$ of various target domains into the CLIP text encoder $E_{txt}$ to obtain the text embedding $\{\mathrm{TrgEmb}^i\}_{i=1}^N = E_{txt}(\{\mathrm{TrgDesc}^i\}_{i=1}^N)$. We utilize the PIN module of PØDA to transform $f_s$ into $\{f_{s \to t}^i\}_{i=1}^N$ which encapsulates the style information of target domains as depicted by the corresponding text descriptions. Specifically, we estimate the distributions of $i$-th target features using the mean $\mu^i$ and standard variance $\sigma^i$ of $f_{s \to t}^i$. We correlate the visual knowledge and the linguistic knowledge by constructing a hybrid cross-modality graph $\mathcal{G}_h$, in which the visual meta-nodes represent the distributions of $\{f_{s \to t}^i\}_{i=1}^N$ and the linguistic nodes represent the text embedding. We define a graph motif that consists of a text embedding $\mathrm{TrgEmb}$ and two extreme features $(P_{v+}, P_{v-})$ on the boundary of a visual feature distribution. The geometry of a motif represents the similarity of cross-modality features and can be utilized for the following alignment process. We mine all motifs $\mathcal{M}$ inherent in $\mathcal{G}_h$ and measure their semantic consistency using our proposed directed edge-based metric. By matching these graph motifs, we achieve feature alignment across domains. Additionally, to prevent the style transformation process from falling into mode-collapse, we introduce directional and contrastive losses.

Our method is different from the previous methods in three points. First, we propose the graph motif-based method designed to estimate and align the semantic consistency between linguistic and visual

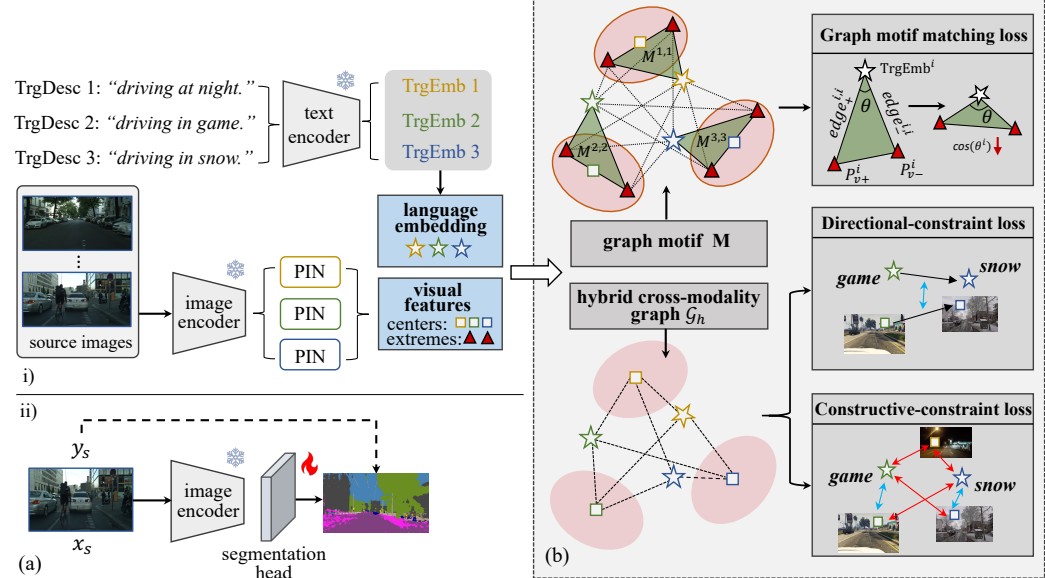

Figure 2: Overview of the proposed zero-shot domain adaptive semantic segmentation framework. (a) shows the two processes of this method: i) stylization of target domain features and ii) target segmenter fine-tuning. (b) is the details of the stylization process. By the optimized PIN modules, we fine-tune the segmentation head of the target segmenter.

features for domain adaptation. Second, we incorporate a directional loss, which establishes a reference system for transforming target features based on the relationships of text embedding. Third, we employ the contrastive loss to equalize the stylistic intensity across different target domains.

## 4.2 MOTIF-BASED FEATURE MATCHING

**Hybrid cross-modality graph.** Given the source image $x_s$ and a pretrained CLIP model, we input $x_s$ into the CLIP image encoder $E_{img}$ to extract their image features $f_s$. For each target domain, we use a text prompt to describe its characteristics, *e.g.*, "driving in snow". We feed all descriptions into the CLIP text encoder $E_{text}$ to obtain the language embedding $\mathbf{P}_l = \{\text{TrgEmb}^i\}_{i=1}^N$ of all target domains. In the zero-shot domain adaptation setting, no data from the target domains is available. Thereby, for the $i$-th target domain, we adopt a text-driven PIN module of PØDA and TrgEmb$^i$ to transform the source features f$_s$ using Eq. (1) and let the transformation result f$_{s \rightarrow t}^i = \text{PIN}^i(\text{f}_s, \mu^i, \sigma^i)$ reflect the style of the target domain. Here, $\{\mu^i\}_{i=1}^N$ and $\{\sigma^i\}_{i=1}^N$ denote the centers and scales of the stylized target image features, respectively. These learnable variables simulate meta-nodes $\mathbf{Q}_v = \{Q_v^i\}_{i=1}^N$, where $Q_v^i$ represents the distribution range of domain-wise visual features. With these meta-nodes of visual features and the linguistic nodes of text prompts, we construct the hybrid graph $\mathcal{G}_h = \{\mathbf{Q}_v, \mathbf{P}_l\}$ to represent the relationships of the features across modalities.

**Graph motif.** To discover the similarity information between visual and linguistic features, we define a triangular graph motif pattern, specifically a third-order subgraph, which recurs within the hybrid graph. Each graph motif is composed of cross-modality feature nodes from various target domains. We mine all graph motifs $\mathcal{M}$ within $\mathcal{G}_h$ and achieve cross-domain feature alignment by measuring their domain-level semantic consistency. For each target domain, we calculate the extreme feature pairs $\mathbf{P}_v = \{(P_{v+}^j, P_{v-}^j)\}_{j=1}^N$ on the boundary of stylized feature distribution range (visual meta-node) $Q_v^j$. The extreme feature pair of $j$-th visual feature distribution is defined as:

$$
\begin{aligned}
P_{v+}^j &= \mu^j + \alpha\sigma^j, \\
P_{v-}^j &= \mu^j - \alpha\sigma^j,
\end{aligned}
\tag{3}
$$

where $\alpha$ denotes the zoom factor used to scale the distribution of visual features. We then connect a language node $\text{TrgEmb}^i \in \mathbf{P}_L$ to an extreme visual feature pair $(P_{v+}^j, P_{v-}^j) \in \mathbf{P}_v$ to form the triangular graph motif $M^{i,j} = \{\text{TrgEmb}^i, P_{v+}^j, P_{v-}^j\} \in \mathcal{M}$. This structured approach allows us to systematically explore and quantify the interactions between the linguistic and visual modalities across different domains.

**Motif matching.** Most existing works focus on increasing the cosine similarity between linguistic and visual features to enhance their semantic consistency. However, these studies often only consider the global features and overlook the diversity inherent in dispersed visual feature distributions. To address this problem, we propose a motif-based method to match the features across modalities more effectively. To align features across domains, we divide the motif set $\mathcal{M}$ into two subsets, *i.e.,* the matched motif set $\mathcal{M}_m$ and the unmatched motif set $\mathcal{M}_{um}$, where $\mathcal{M}_m \cup \mathcal{M}_{um} = \mathcal{M}$ and $\mathcal{M}_m \cap \mathcal{M}_{um} = \emptyset$. A motif $M^{i,j}$ is an element of $\mathcal{M}_m$, iff its linguistic and visual nodes belong to the same domain (*i.e., $i = j$*); otherwise it is an element of $\mathcal{M}_{um}$. Conversely, motifs where the nodes come from different domains (*i.e., $i \neq j$*) are categorized into the unmatched motif set $\mathcal{M}_{um} \subset \mathcal{M}$. Inspired by SOMA Li et al. (2023), we introduce the following metric to estimate the semantic consistency of a graph motif:

$$
\begin{aligned}
sim^{i,j} = 1 - cos(\theta^{i,j}) &= \frac{edge_+^{i,j} \cdot edge_-^{i,j}}{\left\| edge_+^{i,j} \right\| \cdot \left\| edge_-^{i,j} \right\|}, \\
edge_+^{i,j} &= \text{TrgEmb}^i - P_{v+}^j, \\
edge_-^{i,j} &= \text{TrgEmb}^i - P_{v-}^j,
\end{aligned}
\tag{4}
$$

where $edge_+^{i,j}$ and $edge_-^{i,j}$ are two directed edges that originate from the linguistic node and extend towards the extreme visual feature boundaries within the graph motif $M^{i,j}$. $\theta^{i,j}$ denotes the vectorial angle between $edge_+^{i,j}$ and $edge_-^{i,j}$. The similarity measure $sim^{i,j}$, ranging from 0 to 2, quantifies the alignment between the text prompt of the $i$-th target domain and the visual feature distribution of the $j$-th target domain. The matching loss is formulated as follows:

$$
L_{match} = -\sum_{i=1}^{N} \log\left( \frac{\exp(sim^{i,i})}{\sum_{j=1}^{N} \exp(sim^{i,j})} \right).
\tag{5}
$$

By minimizing $L_{match}$, the visual features are enforced to closely align around the corresponding text prompt embedding, thereby enhancing their semantic consistency.

### 4.3 Relationship-constraint Adaptation

**Directional loss.** It is recognized that utilizing a single sample pair to compute cosine similarities can decrease the diversity of the shared feature space and induce mode-collapse Gal et al. (2022). This is particularly problematic in tasks such as image style transformation or feature stylization, where diversity in visual representation is crucial for robust performance. In practice, the source segmenter is often transferred to more than one target domain. Considering that different domains may have unique but related style information in the feature space, one effective approach is to use the directionality of their text descriptions as a reference system. This method maps text description embedding to points in the feature space, where the direction between any two points reflects the stylistic distance or similarity between their corresponding domains. Inspired by Wang et al. (2023); Gal et al. (2022), we apply the following directional loss during the optimization of PIN modules:

$$
L_{dir} = \frac{1}{N^2} \sum_{i=1}^{N} \sum_{j=1}^{N} [1 - cos(\text{TrgEmb}^i - \text{TrgEmb}^j, f_{s \to t}^i - f_{s \to t}^j)].
\tag{6}
$$

The PIN modules will transform the visual features onto a navigation path defined by the directional vectors between text embedding. By applying $L_{dir}$ that penalizes deviations from this path, the model can better maintain diversity in its outputs while adapting to new domains.

**Contrastive loss.** The directional loss provides a reference direction for optimizing target visual features. However, it lacks relational constraints among different domain styles. As observed in Nerf-Art Wang et al. (2023), this can result in uneven degrees of stylization across various domains.

Thereby, we adopt a contrastive learning paradigm to equalize the stylistic intensity across different target domains. This approach is effective in learning discriminative features that are robust across domains with different styles. The contrastive loss is mathematically formulated as:

$$L_{con} = -\frac{1}{N} \sum_{i=1}^{N} \log\left[\frac{\exp(\text{TrgEmb}^i \cdot \text{f}^i_{s\to t})}{\exp(\text{TrgEmb}^i \cdot \text{f}^i_{s\to t}) + \sum_{j\neq i} \exp(\text{TrgEmb}^i \cdot \text{f}^j_{s\to t})}\right], \tag{7}$$

where $\text{f}^i_{s\to t}$ and $\text{f}^j_{s\to t}$ are positive and negative samples, which are different styles of image features related to a specific domain description, $\text{TrgEmb}^i$. This contrastive framework compels the PIN module to differentiate between correct and incorrect style transformations, thereby not only preserving but also enhancing the diversity and accuracy of the style features in the target domain.

## 4.4 OPTIMIZATION

During the target feature stylization process, we employ an overall loss to train the PIN modules, encapsulated as follows:

$$L_{total} = \lambda_{match}L_{match} + \lambda_{dir}L_{dir} + \lambda_{con}L_{con} + L_{PIN}, \tag{8}$$

where $\lambda_{match}$, $\lambda_{dir}$, and $\lambda_{con}$ are the weights of graph motif matching, directional, and contrastive losses, respectively. After training the PIN modules for all target domains, the optimized style parameters $\{\mu^i, \sigma^i\}$ are utilized to transform the source image features into corresponding target features. Subsequently, we fine-tune the target segmenters using the cross-entropy loss $CE(\text{f}_{s\to f}, y_s)$ with stylized features and source labels.

## 5 EXPERIMENTS

### 5.1 DATASETS AND EVALUATION

To assess the efficacy of our proposed method, we conducted several experiments on domain adaptive semantic segmentation tasks. We use the mean Intersection Over Union (mIoU) and mean Pixcel Classification Accuracy (mAcc) to evaluate the segmentation performance of the target segmenters. We compare our method with source CLIP, CLIPStyler, and PØDA. The results of the comparison methods are inherited from Fahes et al. (2023) and Yang et al. (2024). The experiments addressed three types of domain shifts: (a) from clear to adverse weather conditions (Cityscapes → ACDC), (b) from synthetic environments to adverse weather conditions (GTA5 → ACDC), and (c) between real and synthetic environments (Cityscapes ⇌ GTA5). There are three benchmark datasets involved in the experiments: Cityscapes Cordts et al. (2016), ACDC Sakaridis et al. (2021), and GTA5 Richter et al. (2016). Cityscapes is a dataset that contains urban landscapes captured under clear weather conditions. It includes 2,975 training images and 500 validation images, each annotated with 19 pixel-level categories. GTA5 includes 25,000 images rendered using the gaming engine from Grand Theft Auto, also annotated with pixel-level labels. ACDC comprises driving scenes collected under various adverse visual conditions such as fog, nighttime, rain, and snow. It shares the same 19 semantic categories with Cityscapes. We conduct the experiments five times with our proposed method and show the errors of average metrics in the tables.

### 5.2 IMPLEMENTATION DETAILS

We utilize the DeepLabV3+ Chen et al. (2018) architecture to construct both the source and target segmenters, denoted as $G_{s/t}$. In particular, the image encoder $E_{img}$ from the CLIP-ResNet-50 Radford et al. (2021) model serves as the backbone for $G_{s/t}$. During the whole adaptation process, the structure and parameters of $E_{img}$ remain frozen. For initializing the segmentation models, the weights from PØDA Fahes et al. (2023) are used to set up the target segmentation head $H_t$. We employ the Stochastic Gradient Descent (SGD) optimizer Song et al. (2013), with a learning rate of 0.1 and a batch size of 8 over 10,000 iterations to train the PIN modules across all target domains. The zoom factor $\alpha$ for computing $L_{match}$ is set to be 5. The loss weights $\lambda_{match}$, $\lambda_{dir}$, and $\lambda_{con}$ for computing $L_{total}$ are set to 0.1, 0.05, and 0.05, respectively. When fine-tuning the segmentation head of the target segmenter, we adopt the SGD optimizer with a learning rate of 0.01 and a batch size of 8 for 2,500 iterations. All experiments are performed using NVIDIA 3090 GPUs.

Table 1: **Performance comparison of Cistyscapes→ACDC.** In this experiment, the source domain is Cityscapes and the target domains are subsets of ACDC corresponding to four adverse weathers. mIoU and mAcc are average mIoU and average mAcc across all target domains.

| Adaptation | Source2Fog | | Source2Night | | Source2Rain | | Source2Snow | | | |
| Description | driving in fog | | driving at night | | driving under rain | | driving in snow | | mIoU | mAcc |
| Method | mIoU | mAcc | mIoU | mAcc | mIoU | mAcc | mIoU | mAcc | | |
| Source | 49.98 | 65.42 | 18.31 | 34.16 | 38.20 | 58.97 | 39.28 | 54.64 | 36.44 | 53.29 |
| CLIPStyler | 48.87 | 64.31 | 20.83 | 35.32 | 36.97 | 57.46 | 40.31 | 54.42 | 36.75 | 52.87 |
| PØDA | 51.54 | 64.51 | 25.03 | **55.50** | 42.31 | **75.40** | 43.90 | **70.70** | 40.69 | **66.52** |
| ours | **52.71** | **66.38** | **25.11** | 39.83 | **44.20** | 73.86 | **45.20** | 68.40 | **41.80±0.36** | 62.11±0.42 |

Table 2: **Performance comparison of GTA5→ACDC.** In this experiment, the source domain is GTA5 and the target domains are subsets of ACDC corresponding to four adverse weathers. mIoU and mAcc are average mIoU and average mAcc of all target domains.

| Adaptation | Source2Fog | | Source2Night | | Source2Rain | | Source2Snow | | | |
| Description | driving in fog | | driving at night | | driving under rain | | driving in snow | | mIoU | mAcc |
| Method | mIoU | mAcc | mIoU | mAcc | mIoU | mAcc | mIoU | mAcc | | |
| Source | 33.20 | 42.51 | 12.22 | 22.56 | 33.32 | 43.15 | 32.33 | 40.60 | 27.76 | 37.20 |
| CLIPStyler | 30.79 | 40.37 | 11.12 | 20.18 | 31.17 | 40.06 | 30.65 | 38.97 | 25.93 | 34.89 |
| PØDA | 35.76 | 44.98 | 13.35 | 25.24 | 34.19 | 45.93 | 33.81 | 42.10 | 29.27 | 39.56 |
| ours | **36.47** | **45.89** | **16.44** | **30.42** | **35.33** | **46.03** | **34.56** | **43.43** | **30.70±0.29** | **41.44±0.40** |

Table 3: **Performance comparison between Cistyscapes and GTA5.** CS→GTA5 and GTA5→CS represent the the adaptation tasks of Cistyscapes→GTA5 and GTA5→Cistyscapes, respectively. *styler* represents the CLIPStyler method.

| Task | Method | road | sidewalk | building | wall | fence | pole | traffic light | traffic sign | vegetation | terrain | sky | person | rider | car | truck | bus | train | motorcycle | bicycle | mIoU |
|---|---|---|---|---|---|---|---|---|---|---|---|---|---|---|---|---|---|---|---|---|---|
| | | | | | | | Description = "driving in a game" | | | | | | | | | | | | | | |
| CS→GTA5 | source | 68.7 | 22.7 | 78.8 | 36.8 | **17.3** | **39.7** | 39.3 | 14.8 | 72.6 | 22.5 | 87.3 | 57.5 | 26.1 | 74.3 | **44.6** | **20.5** | 0.0 | 18.3 | 10.4 | 39.6 |
| | styler | 73.1 | **29.9** | 77.9 | 25.5 | 11.7 | 39.7 | 35.9 | **24.0** | 67.4 | 12.8 | 88.8 | 46.6 | 33.4 | 72.0 | 42.8 | 11.1 | 0.0 | 28.8 | 14.6 | 38.7±0.16 |
| | PØDA | 73.9 | 22.7 | 78.8 | 37.5 | 14.2 | 37.0 | 33.1 | 17.3 | 72.4 | 26.2 | **88.9** | **62.7** | 37.0 | 74.3 | 43.0 | 11.9 | 0.0 | 35.3 | 13.9 | 41.1±0.48 |
| | ours | **75.6** | 24.4 | **79.6** | **37.9** | 14.6 | 38.6 | **39.8** | 21.8 | **73.8** | **30.2** | 88.7 | 61.8 | **40.6** | **75.4** | 43.6 | 12.9 | 0.0 | **37.6** | **17.4** | **42.9±0.29** |
| | | | | | | | Description = "driving in real" | | | | | | | | | | | | | | |
| GTA5→CS | source | 59.0 | 20.9 | 72.8 | 16.5 | **24.6** | 31.4 | 34.8 | 23.6 | 82.1 | 17.0 | 66.3 | **63.5** | 14.7 | **81.3** | **20.8** | 17.2 | 4.7 | **20.6** | 19.6 | 36.4 |
| | styler | 66.7 | 23.6 | 64.1 | 5.1 | 3.7 | 20.7 | 19.3 | 18.1 | 81.7 | 12.4 | **81.0** | 54.6 | 0.5 | 73.5 | 20.7 | 22.3 | 4.0 | 15.8 | 10.7 | 31.5±0.21 |
| | PØDA | 84.3 | **36.7** | 79.4 | 18.3 | 16.5 | **36.9** | **38.5** | 33.8 | 82.4 | 19.1 | 75.9 | 62.7 | 16.5 | 75.5 | 15.7 | 19.6 | **11.3** | 16.5 | **21.8** | 40.1±0.52 |
| | ours | **86.1** | 35.5 | **80.3** | **18.4** | 18.8 | 36.8 | 37.3 | 29.0 | **83.6** | **19.4** | 77.6 | 63.4 | **16.5** | 79.3 | 19.6 | **25.6** | 5.8 | 19.1 | 21.0 | **40.7±0.38** |

## 5.3 COMPARISON WITH STATE-OF-THE-ARTS

**Cityscapes→ACDC.** Tab. 1 shows the experimental results. Our method achieves the average mIoU and mean accuracies of 41.8 and 62.11, the average mIoU perforamnce exceeding all existing methods to achieve sota performance. In comparison with the source CLIP model, we achieve the average mIoU and mean accuracies gains of 5.36 and 8.82, respectively. Compared with the Unet-based CLIPStyler, our method surpasses it by 5.05 and 9.24 in average mIoU and mean accuracies across four target subsets, respectively. Additionally, our method demonstrates improvements of 1.11 in average mIoU over the closely related method PØDA. The experimental results show that our method can effectively transfer CLIP from clear weather data to adverse weather data.

**GTA5→ACDC.** Tab. 2 presents the comparison results. Our method achieves a 30.70 average mIoU and a 41.44 average mean accuracies, exceeding all existing methods. When compared with CLIP-Stlyer, our method outperforms it by 4.77 in average mIoU and 6.55 in average mean accuracies, respectively. We also compare our method with the closely related PØDA, our method surpasses it by 1.43 and 1.88 in average mIoU and average mean accuracies, respectively. The experimental

Table 4: **Comparison results of different components of total loss.** $\overline{\text{mIoU}}_{C2A}$ and $\overline{\text{mIoU}}_{G2A}$ denote the average mIoUs of Cityscapes→ACDC and GTA5→ACDC.

| $L_{match}$ | $L_{dir}$ | $L_{cons}$ | $\overline{\text{mIoU}}_{C2A}$ | $\overline{\text{mIoU}}_{G2A}$ |
|:---:|:---:|:---:|:---:|:---:|
| | | | 40.69 | 29.27 |
| ✓ | | | 41.31 | 30.45 |
| | ✓ | | 40.85 | 29.86 |
| | | ✓ | 41.12 | 29.90 |
| | ✓ | ✓ | 41.37 | 30.11 |
| ✓ | ✓ | | 41.59 | 30.53 |
| ✓ | | ✓ | 41.68 | 30.49 |
| ✓ | ✓ | ✓ | **41.80** | **30.70** |

results show that our method can effectively transfer CLIP from synthetic data to adverse weather data.

**Cityscapes⇌GTA5.** Tab. 3 lists the detailed category-wise comparison results. Our method achieves 42.9 and 40.7 mIoUs for two tasks, surpassing the source CLIP model by margins of 3.3 and 4.3. In comparison with CLIPStyler, our method improves the average mIoUs by margins of 4.2 and 9.2. We also compare our method with the closely related PØDA, our method achieves improvements of 1.8 and 0.6 mIoUs. The experimental results show that our method can effectively transfer CLIP between real data and synthetic data. Therefore, our method has potential for applications in autonomous driving, such as using synthetic data to enrich datasets for improving the performance of segmentation models in the real world.

### 5.4 ABLATION STUDIES

To evaluate the effectiveness of our proposed method, we present the following ablation studies in Tab. 4 and Tab. 5. The experiments are conducted on the adaptation tasks from clear weather to adverse weathers (Cityscapes→ACDC) and from synthetic data to adverse weathers (GTA5→ACDC).

**Ablation on the effectiveness of each component.** Tab. 4 lists the average mIoU performance of our method using different loss components. Take Cityscapes→ACDC for example, each component of $L_{total}$ contributes significantly to the performance of target segmenters. Among these components, the graph motif matching loss has the most substantial impact, enhancing the average mIoU by 0.62 from 40.69 of the baseline method. The addition of directional and contrastive losses improves the segmentation performance by 0.16 and 0.43, respectively. When using both relationship-constraint losses simultaneously, the average mIoU increases from 40.69 to 41.37. By integrating the motif matching loss and two relationship-constraint losses, our method achieves improvements in average mIoU by 0.90 and 0.99, respectively. When using all loss components, our method achieves the best average mIoU of 41.80 and 30.70 for Cityscapes→ACDC and GTA5→ACDC, respectively.

**Ablation on the motif zoom factor.** Tab. 5 shows the impact of different motif zoom factors. We set $\alpha$ to 1, 3, 5, 7, and 9 and carried out comparative experiments to assess the changes in segmentation performance. The results indicate that motif matching does not significantly improve segmentation performance when a is less than 5. This limited effectiveness can be attributed to the smaller $\alpha$ values resulting in a too compact range for the visual meta-node in feature space. Such compactness affects the separability of the language-vision directed edges. Conversely, a large $\alpha$ causes confusion within the visual meta-node due to overly expanded feature spaces, which can blur the distinctions necessary for effective segmentation. The optimal $\alpha$ is experimentally determined to be 5 for all tasks.

### 5.5 QUALITATIVE RESULTS

**Result comparison.** Fig. 3 shows the semantic segmentation qualitative results for the adaptation task from Cityscapes to ACDC. Compared to PØDA, our method demonstrates a more precise segmentation results. This improvement is particularly noticeable in the segmentation of large areas such as the sky and road, where our approach has achieved significant performance enhancements.

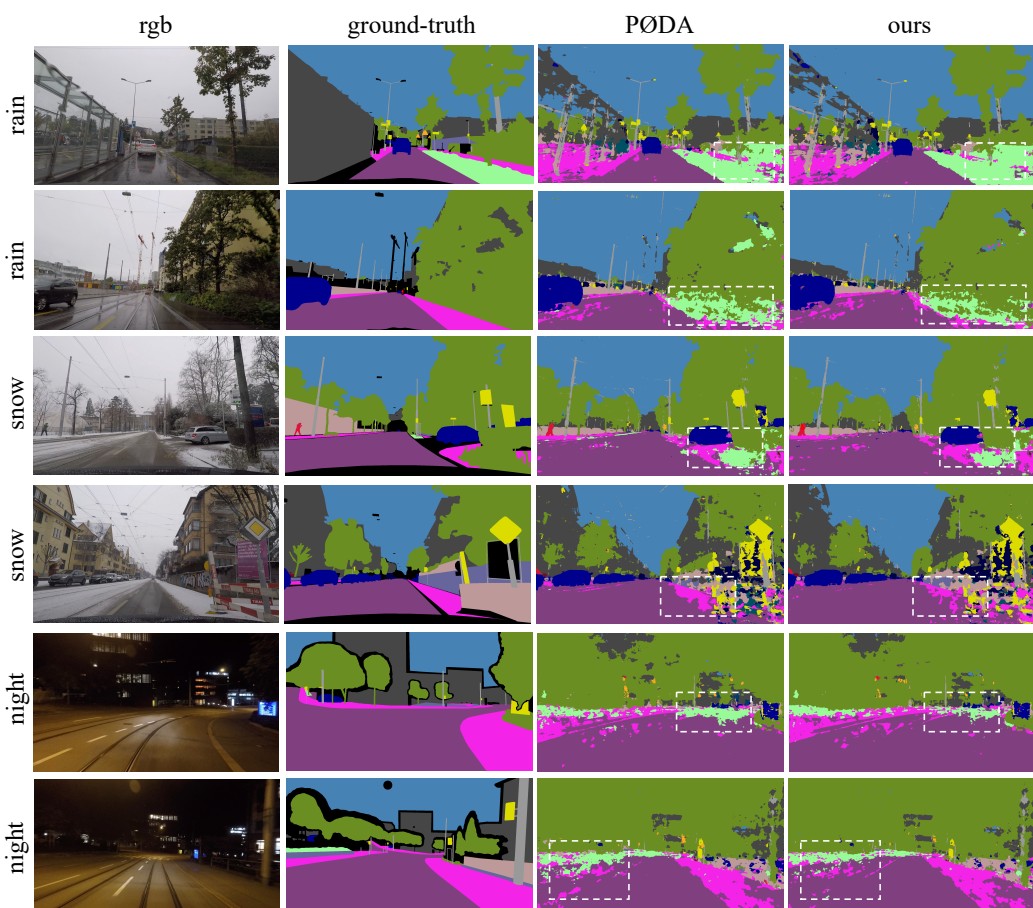

Figure 3: Qualitative comparison results on the task of Cityscapes→ACDC.

Table 5: **Comparison results of different motif matching zoom factors.** $\overline{\text{mIoU}}_{C2A}$ and $\overline{\text{mIoU}}_{G2A}$ denote the average mIoUs of Cityscapes→ACDC and GTA5→ACDC. The larger the value of $\alpha$, the wider the distribution range of visual meta-nodes in the graph motif, resulting in greater separation of the extreme values of visual features.

| $\alpha$ | 1 | 3 | 5 | 7 | 10 |
|---|---|---|---|---|---|
| $\overline{\text{mIoU}}_{C2A}$ | 40.88 | 41.47 | **41.80** | 41.63 | 41.52 |
| $\overline{\text{mIoU}}_{G2A}$ | 29.61 | 30.13 | **30.70** | 30.55 | 30.49 |

## 6 CONCLUSION

We propose a graph motif-based adaptation method to deal with the zero-shot domain adaptive semantic segmentation problem. We employ the CLIP encoders to extract the visual and linguistic features and adopt the prompt-driven instance normalization module to transform the source features into stylized target features. We propose a graph motif structure to represent the relationships among the visual feature distributions and text embedding. By reducing the language-vision directed edges in the motifs, we pull visual features to the text embedding centers of target domains. In addition, we employ the relationship-constraint losses, *i.e.,* directional and contrastive losses, to stabilize the learning process and improve the robustness of the adaptation. The comprehensive experiments verify the effectiveness of our proposed method.

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
