# OpenReview forum: "Text-driven Zero-shot Domain Adaptation with Cross-modality Graph Motif Matching"
_ICLR.cc/2025/Conference — ICLR 2025 Conference Withdrawn Submission_

### Official Review · Reviewer_c3v1 · 2024-10-30

**Soundness:** 2
**Presentation:** 3
**Contribution:** 2
**Rating:** 3
**Confidence:** 4

**Summary:**

The paper introduces a zero-shot domain adaptive segmentation method that aligns source and target features using graph motifs. By improving cross-modality feature alignment and applying constraint losses, the approach enhances robustness and prevents mode collapse. Experimental results validate its effectiveness.

**Strengths:**

1. The paper is overall well-written and easy to follow.
2. The proposed method is overall simple and easy to understand.
3. Extensive experiments across multiple datasets are sufficient.

**Weaknesses:**

Major：

1. Inferior performance. Comparing Table 1 in this paper with Table 2 in Ref A and Table 3 in Ref B, the performance is significantly lower. For example, in terms of  mAcc, this paper reports 66.38 in Table 1, while Ref A reports 80.2 in its Table 2. What could be the potential reasons for the performance gap?

2. No computational cost analysis. The authors claim the existing methods cannot accomplish a proper balance about the computation cost. However, there are no quantitative comparisons provided to demonstrate the superiority of this paper in computational cost. Could you provide specific metrics (e.g., training time, inference time, memory usage) to quantitatively compare the computational efficiency of your method to existing approaches？

3. No ablation analysis of the cosine similarities. The authors claim the cosine similarities of each paired image and text sample can cause the shared feature space to lack diversity. However, there's a lack of relevant ablation analysis. Could you compare your method against other baselines using only cosine similarities and demonstrate how this impacts feature diversity?

4. The related work section needs updating. For instance, there are no papers published in 2024 included in the Vision-Language Models section, and only one in the Unsupervised Domain Adaptation section. Additionally, Ref B is highly relevant but is neither cited nor analyzed.

Ref A: Unified Language-driven Zero-shot Domain Adaptation. In CVPR2024 (pp. 23407-23415).

Ref B: Collaborating Foundation Models for Domain Generalized Semantic Segmentation.  In CVPR2024 (pp. 3108-3119).

Minor:
1. PØDA should be cited upon its first mention in this paper.

**Questions:**

Please see the above weaknesses.

---

### Official Review · Reviewer_ryhg · 2024-11-01

**Soundness:** 2
**Presentation:** 1
**Contribution:** 2
**Rating:** 3
**Confidence:** 3

**Summary:**

This paper proposes a graph motif-based method to address the zero-shot domain adaptive semantic adaptation (ZSDA-SS) problem. First, it constructs motif structures to pull visual features toward the language feature center, aligning the two modalities. Then, it employs directional and contrastive losses to help stabilize the learning process.

**Strengths:**

This paper propose matching graph motifs across modalities and domains to enhance alignment of cross-modality features and diversity.

**Weaknesses:**

1. The novelty of the relationship-constraint adaptation is very limited. Directional loss and contrastive loss are present in many existing works, and the ablation experiment (Table 4) shows that these two components are not effective. For example, comparing the last line with the second-to-last line illustrates this point.

2. Regarding Figure 1, the information presented (e.g., star, triangle) is not well explained; however, it is clarified in Figure 2, which does not align with typical reading habits. Additionally, Figure 1 is missing a component between the image encoder and the stylized target feature, which I believe is the PIN component.

3. In Figure 2, the meaning of the red and blue rows is not clear.

4. In motif matching, there appears to be an error in Formula 4, as it is missing a '1 - '.

5. From lines 276 to 285, it is unclear whether motif matching is applied to matched motifs or all motifs.

6. In the experiment (Table 1), it appears that there is only minimal improvement, with some negative results when comparing the proposed method to PØDA.

**Questions:**

See weaknesses

---

### Official Review · Reviewer_XciE · 2024-11-03

**Soundness:** 2
**Presentation:** 2
**Contribution:** 2
**Rating:** 3
**Confidence:** 4

**Summary:**

This paper tackles the problem of text-driven zero-shot domain adaptation in semantic segmentation. The authors adopt the prompt-driven instance normalization (PIN) module from PODA [1], which is an instance normalization used to augment styles of visual features using text descriptions. The authors propose to optimize PIN using 3 additional losses. The first is a motif matching loss, where triangular graph motifs are constructed from a language node (target description embedding) and an extreme visual feature pair computed from centers and scales of stylized features. The idea is to enforce visual features to align around the text embedding describing the corresponding domain. The second loss is a directional loss aiming at matching directions between two embeddings in each of the text and visual space instead of direct matching. The third loss is a contrastive loss that pulls text embedding and corresponding visual features together and pushes apart dissimilar ones. The method is tested on different semantic segmentation settings: Cityscapes $\rightarrow$ ACDC, GTA5 $\rightarrow$ ACDC and  Cityscapes$\rightleftharpoons$ GTA5 and is shown to outperform the most relevant baseline PODA [1].


[1] Fahes et al., Poda: Prompt-driven zero-shot domain adaptation. ICCV 2023

**Strengths:**

* The idea of representing as a cross-modality graph the visual styles and text embeddings is interesting as approach to model the interaction among them.
* The graph motif matching loss is interesting and is shown experimentally to improve the performance of PODA [1].
* The paper is well-illustrated.

**Weaknesses:**

* The method seems to be computationally more expensive than [1]. What are the additional time and memory constraints that are implied by the ensemble of the proposed components (constructing the hybrid cross-modality graph, calculating extreme features, mining graph motifs, optimizing PIN ...)? The additional losses improve the mIoU in general, however I am wondering how scalable and practically applicable is the approach, especially that the segmentation performance improvement is not very high.
* How are the values of $\lambda_{match}$, $\lambda_{dir}$ and $\lambda_{con}$ selected? Looking at the ablation on these values in the appendix (Table 6), some combinations lead to lower performance than [1], which raises doubts about practical selection of these hyperparameters when no target data is available.
* The authors of [2] propose also an extension of [1]. Is there any reason the authors do not compare with [2]? [2] is advantageous in training a single segmentation network for all the unseen domains. Are the contributions of this paper complementary to those of [2]?
* The authors of [A] use the same architecture (ResNet-50, DeepLabV3+) and less constrained language information (random texts instead of target-specific texts), and show strong results on domain generalization. This calls into question the practical interest of extending PODA [1] in the zero-shot adaptation setting. One interesting way to better assess the merit of the proposed method is to show if it exhibits some complementarity with [A].
* The image encoder being frozen during training, how would the method perform with better segmentation heads like Mask2Former [B] for instance? Freezing the backbone appears to be a better choice with strong segmentation heads [C]. In this case, would using PIN with the same losses have the same effectiveness?
* The qualitative results do not reflect significant improvement, which are hard to see even in the added boxes.
* The authors mention that coarsely aligning compressed visual features leads to the loss of domain-specific information. Any evidence about this? It is true that the alignment is performed in the latent space, however the optimizable variables in PIN [1] are affine transformations of features having spacial dimensions (height and width). This makes the claim a bit strong and not well-supported by evidence.

[2] Yang et al., Unified Language-driven Zero-shot Domain Adaptation. CVPR 2024
[A] Fahes et al., A Simple Recipe for Language-guided Domain Generalized Segmentation. CVPR 2024
[B] Cheng et al., Masked-attention Mask Transformer for Universal Image Segmentation. CVPR 2022
[C] Wei et al., Stronger, Fewer, & Superior: Harnessing Vision Foundation Models for Domain Generalized Semantic Segmentation. CVPR 2024

**Questions:**

* Is there any interpretation why the mAcc is less than PODA while the mIoU is higher in Tab.1? For instance, in Source2Night case mAcc is ~15% higher for PODA while the mIoU is only 0.08% higher for the proposed method.
* "Mode-collapse" phenomenon is mentioned many times in the paper to motivate the utilization of directional and contrastive losses. Is there any evidence of mode-collapse when these losses are not used? Especially that the performance is marginally harmed when not using these losses.
* Which features are stylized with PIN? Is it shallow or deep features?
* Line 166: It is not clear to me how average pooling operation is used. Shouldn't the features be passed through the image encoder, and at the end it is the attention pooling layer of CLIP that is applied?
* PIN is trained for 10,000 iterations while in PODA[1] it is trained for 100 iterations. Could the authors comment on this difference please?


Potential typos:
* Line 011: semantic adaptation $\rightarrow$ semantic segmentation.
* Line 066: "under the driven" $\rightarrow$ potential typo.
* Line 148: stylization of target domain features $\rightarrow$ stylization of source domain features.
* Line 161: similarity between $f_{s \rightarrow t}$ and CLIP text embedding $\rightarrow$ similarity between $f_{s \rightarrow t}$ embedding and CLIP text embedding.
* Line 199, 237 : Stylization of target domains $\rightarrow$ Stylization of source domains?
* Line 344: $f_{s \rightarrow f}$ $\rightarrow$ $f_{s \rightarrow t}$?
* Line 353: Pixcel $\rightarrow$ Pixel?
* Line 421: perforamnce $\rightarrow$ performance
* Line 473: 1,3,5,7,9 $\rightarrow$ 1,3,5,7,10? (not matching Table 5).
* Line 475: when a is less than 5 $\rightarrow$ when $\alpha$ is less than 5?

---

### Note · Authors · 2024-11-13

I have read and agree with the venue's withdrawal policy on behalf of myself and my co-authors.